# OpenReview forum: "VideoJail: Exploiting Video-Modality Vulnerabilities for Jailbreak Attacks on Multimodal Large Language Models"
_ICLR.cc/2025/Workshop/BuildingTrust — BuildingTrust_

### Official Review · Reviewer_rdPb · 2025-02-16
**Review of the paper**

**Rating:** 7
**Confidence:** 4

**Review:**

## Summary
This paper investigates the security risks associated with the use of video inputs in multimodal large language models (MLLMs), an area that has not been thoroughly explored in prior research. Through detailed empirical analysis, the paper highlights how the inclusion of video data creates specific vulnerabilities that can be exploited in jailbreak attacks. The authors then introduce VideoJail, an innovative attack method that utilizes video generation models to enhance the harmful effects of static images, guiding the model's focus toward malicious queries embedded within the video content. To address the defenses in closed-source MLLMs, which are designed to detect harmful content, the paper also presents an enhanced version, VideoJail-Pro. This advanced approach takes advantage of the temporal and dynamic aspects of video, employing a jigsaw-like frame rearrangement technique to successfully bypass safety mechanisms and improve attack efficacy.

## Strengths
**1. The topic is crucial and unexplored:** With the rapid development of MLLMs, safety issues have become increasingly important. The paper analyzes the impact of different types of inputs on the toxicity of MLLM outputs, providing a comprehensive viewpoint in this field.

**2. The attack method is simple yet effective:** This paper presents two jailbreak attack methods: VideoJail and VideoJail-Pro. Both novel methods are not complicated but have proven to be quite effective through experiments.

**3. The experiments are comprehensive and persuasive:** The paper provides numerous experiments on different types of state-of-the-art (SOTA) MLLMs, including both open-source and closed-source models, making the results more reliable and persuasive.

## Weaknesses
**1. Deeper analysis of experimental results:** Although the article mentions the attack success rate (ASR), a deeper discussion of why these results occurred would be helpful. More details on the models’ responses and why certain models are more vulnerable to video inputs, or why increasing model parameters doesn’t always improve defenses, could add more value.

**2. Discussion of security measures:** While VideoJail-Pro's attack method is discussed, it would be beneficial to further explore how to strengthen defenses against such attacks. This would not only focus on the attack itself but also provide suggestions for future improvements, making the article more constructive.

---

### Official Review · Reviewer_A961 · 2025-03-04
**Strong video jailbreaking results and analysis on multimodal language models**

**Rating:** 7
**Confidence:** 3

**Review:**

### Summary

This work conducts an empirical analysis on the vulnerabilities of multimodal LLMs in the context of video, and demonstrates the vulnerabilities introduced with the inclusion of this modality. They propose *VideoJail* (and an extension to it) which in a black-box threat model, can successfully jailbreak most open-sourced MLLMs, as well as closed-sourced (to a somewhat lesser extent).

### Strengths

- Strong results between VideoJail and VideoJailPro across all models tested
- The approach of tiling up the inputs in the “jigsaw game” for VideoJailPro is interesting
- Comprehensive evaluation of a variety of MLLMs w.r.t. their vulnerability to video-based jailbreak attacks, across a variety of jailbreak input formats
- Well written, clear, and easy to follow

### Weaknesses

- The approach seems very sensitive to the formatting of the text prompts, but it does seem to consistently work across a variety of MLLMs; this is an interesting finding but perhaps could be a brittle point of the attack algorithm

### Questions/Comments

- Most models (other than Qwen2-VL-72B) seem to have comparable ASRs between $I_{typ} + T_{vidjail}$ and  $V_{typ} + T_{vidjail}$; if I understood correctly, $I_{typ}$  was a static input image which embeds the harmful text; as I’m not as familiar with safety research in the context of vision/language models, are non-video VLLMs similarly vulnerable to comperably simple attacks? Or does the inclusion of video as an input modality open up the attack surface in a way that makes the model more brittle to these sorts of attacks? Is this something we could expect to improve with better (safety) training, potentially as we get more “community sourced red-teaming” efforts?

---

### Official Review · Reviewer_6QMf · 2025-03-04
**Review of "VideoJail: Exploiting Video-Modality Vulnerabilities for Jailbreak Attacks on Multimodal Large Language Models"**

**Rating:** 8
**Confidence:** 4

**Review:**

### **Paper summary**

This paper focuses on the role of the video modality in multimodal large language models (MLLMs) with respect to adversarial attacks, specifically jailbreaks. The authors’ contributions are twofold: (1) they conduct a systematic analysis of the video modality’s influence in the harmlessness performance of MLLMs, and (2) propose a novel jailbreak method for MLLMs, VideoJail, that uses a combination of typographic image creation, video generation, a tailored adversarial text prompt, and for closed-source MLLMs, an additional image processing step to bypass built-in alignment mechanisms. Their analysis demonstrates unique and significant vulnerabilities in the video modality, and the strong ability for VideoJail to exploit these vulnerabilities (>90% ASR on both open-source and proprietary MLLMs).

### **Reasons to accept**

1.	This paper demonstrates novelty in exploration of jailbreak vulnerabilities and crafting jailbreak methods that are specific to the video modality (and non-trivial when extending from the image modality).
2.	The empirical study of the role of the video modality in MLLM jailbreak attacks is insightful, and thorough in its exploration of model architecture, video framerate, model size, and comparison to the image modality.
3.	The evaluation of VideoJail, encompassing various model architectures, safety dimensions, baselines, and MLLM types (open vs closed source) is comprehensive and sufficient.

### **Reasons to reject**

(none)

### **Comments for authors**

1.	Overall, this is a very strong, well motivated, and well organized paper. The remainder of my comments consist of requests for clarifications or suggestions for readability.
2.	For the calculation of ASR, would it be possible to clarify the substring lookup and GPT-3.5 evaluation are aggregated / used together to compute the final ASR?
3.	Why was 2 seconds chosen for the video length? Would the video length matter in the same way that the framerate (Figure 4b) does for varying the ASR?
4.	For VideoJail-Pro, are the four frames (line 411) the only four frames in the video? How does this relate to / be consistent with the 2 second video length and varying frames per second described in Section 2?
5.	For VideoJail-Pro, is it possible to present baseline ASRs for the closed-source MLLMs (as in Table 1)? This would illustrate the change in ASR in addition to the results in Figure 8b.
6.	Typos on the following lines: 38, 235, 250
7.	T_fig (first used in Equation 1, line 255) is not defined earlier in the paper.
8.	T_vidjail (introduced in lines 307-309) should be defined in the text (rather than Figure 5) to enhance readability.
9.	The term Q is defined on 3 different equations; I suggest making them distinct terms to avoid potential confusion in notation.
10.	I suggest changing “four experimental setups” (line 324) to “five experimental setups” to be consistent with the number of rows per model in Table 1.

---

### Decision · Program_Chairs · 2025-03-04

**Decision:**

Accept

**Comment:**

The reviewers agree that the topic is important and underexplored, making this paper a strong fit for the workshop. The paper is praised for its comprehensive empirical evaluation, clear presentation, and novel attack techniques. Areas for improvement include a deeper analysis of why models are vulnerable to video-based attacks and a discussion on potential defenses to make the work more constructive. Overall, the paper addresses a hard and crucial problem in MLLM security, making it a valuable contribution to AI safety research.